# Modeling and Calibration of Pressure-Sensing Insoles via a New Plenum-Based Chamber

**DOI:** 10.3390/s23094501

**Published:** 2023-05-05

**Authors:** Italo Belli, Ines Sorrentino, Simeone Dussoni, Gianluca Milani, Lorenzo Rapetti, Yeshasvi Tirupachuri, Enrico Valli, Punith Reddy Vanteddu, Marco Maggiali, Daniele Pucci

**Affiliations:** 1Artificial and Mechanical Intelligence Research Line, Istituto Italiano di Tecnologia (IIT), Center for Robotics and Intelligent Systems, 16163 Genova, Italy; 2Machine Learning and Optimisation, The University of Manchester, Manchester M13 9PL, UK; 3iCub Tech Facility, Italiano di Tecnologia (IIT), Center for Robotics and Intelligent Systems, 16163 Genova, Italy

**Keywords:** wearable sensors, sensor calibration, model identification, sensorized insoles, capacitive sensors

## Abstract

This paper proposes a novel method to reliably calibrate a pair of sensorized insoles utilizing an array of capacitive tactile pixels (*taxels*). A new calibration setup is introduced that is scalable and suitable for multiple kinds of wearable sensors and a procedure for the simultaneous calibration of each of the sensors in the insoles is presented. The calibration relies on a two-step optimization algorithm that, firstly, enables determination of a relevant set of mathematical models based on the instantaneous measurement of the taxels alone, and, then, expands these models to include the relevant portion of the time history of the system. By comparing the resulting models with our previous work on the same hardware, we demonstrate the effectiveness of the novel method both in terms of increased ability to cope with the non-linear characteristics of the sensors and increased pressure ranges achieved during the experiments performed.

## 1. Introduction

The rise of AI techniques to process large amounts of data has provided solutions to complex and long-standing problems [1,2]. When attempting to apply machine learning to predict human quantities, however, one of the main problems is the need to acquire large amounts of data to train the underlying AI algorithms. In this context, wearable sensors represent attractive devices for the retrieval of data from humans, both in the laboratory and during everyday activities, with the potential to enable application of AI in several market areas, such as medical rehabilitation, human-robot collaboration and sport activity monitoring [3,4,5]. Sensorized insoles have significant potential as wearable devices since they could replace more traditional force plates, being more portable and able to provide a large amount of data for real-time analysis of human gait and feet pressure distribution. This paper aims to advance the reliable use of wearable sensors by presenting an enhanced calibration setup (see Figure 1) and methods for capacitive tactile sensors applied to lightweight wearable insoles introduced in [6].

### 1.1. Existing Insoles

In recent years, many sensorized wearable insoles have been developed and the literature describing them is vast and varied. Although many of these devices are still at the experimental stage, some can already be found on the market, creating pressures for increased research to meet ever-growing customer expectations. Among the commercially available insoles, Adidas GMR™ consists of a foam insole with a sensorized tag inserted under it, MoveSole^®^ features seven sensors per foot [7], and PODOSmart^®^ contains an inertial platform [8]. This brief list, though not exhaustive, highlights the low number of sensors that are usually integrated in such devices (to the best of our knowledge, only XSENSOR^®^ currently offers an insole using a much higher number of sensing units). With respect to published results, [9] presented insoles that were based on 75 resistive sensors each, but the measurement frequency was limited to just 13 Hz. In [10], interesting results were achieved, however, only 24 sensors were used and the range of pressure considered (0–2 bar) did not cover typical working conditions experienced by the sole in many situations (e.g., when running). From the point of view of gait analysis, many other investigations have used a limited number of sensors that, while enabling recognition of discrete gait events and extraction of meaningful features, do not provide information that would enable a full understanding of foot-ground interactions and their continuous real-time variations to be obtained [11,12,13,14]. To address this concern, ref. [6] presented an insole carrying as many as 280 capacitive pressure sensors. This hardware, and a more refined calibration pipeline for it, is the focus of this investigation.

### 1.2. Existing Calibration Setups

Calibration is a crucial step in the delivery of meaningful sensor data and the setup used at this stage must be designed accurately. In the following, we present a selection of different calibration setups that have been used in the context of wearable sensors, in particular, where pressure sensors have been employed. In [15,16,17], sensors were calibrated via dynamic loading (performed with a load cell). While quite attractive, the methodology used is unsuitable when dealing with a large number of sensors. A similar problem affected the investigation reported in [18], where each sensor’s strain was estimated by combining measurements coming from a webcam together with the variation in its electrical resistance. In relation to sensorized insoles, another popular approach has been to place them on force plates and use the measurements obtained as a ground truth reference for the calibration procedure, as in [9]. This approach requires the use of expensive hardware and a cheaper version was proposed in [19]. However, these methods rely on third-party devices that could introduce biases and can only be used for planar sensor arrays. To solve these problems, in [6,20,21] the same sensors considered here were calibrated by inserting them into an air-tight bag and using a pump to decrease the internal pressure in the bag. This induced a pressure difference between the inner part of the bag (where the sensors were) and the outside, effectively exciting all the taxels homogeneously due to the isotropic characteristic of pressure. This approach is very appealing, since it can be readily used with non-flat sensor arrays, though the maximum pressure that can be reached is constrained to be 1 bar at maximum. In light of this, ref. [6] considered force plate references as a second input to the calibration algorithm. To overcome this limitation, in Section 3, a different and more versatile setup is proposed.

### 1.3. Overview of Models Used for Calibration

The setups described above can be used to acquire insights into how the electrical output of the sensors varies given the evolution of the input variables (i.e., the force or pressure). If the physics behind the sensor were completely known, this procedure would be straightforward; however, such a white-box approach usually results in heavy simplifications of the model, as in [22]. Since the real behaviour of the sensor is complex, and includes non-linearities (e.g., hysteresis), its input-output characteristic is frequently found via different approaches. Deep learning techniques were used in [15,18,23], to more accurately capture descriptions of the sensors. However, the choice of the architecture used is often difficult to justify as it is difficult to interpret the physical meaning of the final model obtained. In [9], an exponential relationship between the applied force and the measured resistance was found, while [10] identified a simple linear relationship between capacitance variation and applied pressure, but repeatability and accuracy were a result mostly of the hardware chosen. Polynomial models appear to be promising and have been used in many studies [6,13,20,21]. Usually, the sensor response is modeled as depending on just the input at a single time step (disregarding the previous history of the system), and the choice of the polynomial order itself is based on the sensor behaviour, but without extensive exploration or justification.

### 1.4. Contribution

In view of the above, this paper presents an enhanced calibration procedure for sensor-dense wearable insoles, using a new, more versatile, yet simple, calibration setup. We aim to contribute to the state of the art in three ways:by introducing a novel, pressure-based, calibration setup;by proposing a refined calibration procedure for our insole, using a polynomial model of each sensor that also considers information coming from its past outputs;by validating the model found with static and dynamic trials performed using our setup.

The remainder of the paper is organized as follows: Section 2 contains background information for the reader in terms of the hardware used and the underpinning physics. Section 3 provides details of our innovative calibration setup. Section 4 presents our sensor modeling and the identification procedure. Section 5 presents the results. Section 6 draws our conclusions and provides suggestions for future work.

## 2. Background

We present below the working principles of the sensing technology contained in the wearable insoles and consider the formal relationships between the physical quantities on which the sensor measurements are based. Lastly, a synthetic formulation of the problem tackled here is provided.

### 2.1. Sensing Technology

A thorough description of our wearable insole is presented in [6]. For convenience, we briefly recall its main features here. The core sensing functionality is achieved with an array of capacitive sensors that is placed on top of a 3D-printed plastic insole. Each wearable insole contains 280 individual tactile pixels. Using the same technology as in the skin of the humanoid robot iCub [24], the taxels are distributed in triangular patches. Each triangle carries an AD7147 analog-to-digital converter, capable of single electrode capacitive sensing and connected to 10 taxels (+2 extra taxels for temperature compensation). The analog values are discretized using eight bits, instead of the 16 bits that the AD7147 offers, to reduce the noise of the measurements, which are then sent to a host PC at a rate of 50 Hz through a CAN bus interface using YARP middleware [25].

A layer of dielectric material is placed on the top part of the insoles and is covered by a conductive sheet (non-woven coppered PET) to permit the creation of an array of capacitors in which one plate is connected to the ADC converter and the other plate to the conductive sheet itself. To further reduce the measurement noise, both the ADC converters and the conductive sheet are commonly grounded. The dielectric material chosen for our experiment is Infinergy^®^ from BASF (1 cm thick), whose properties of very low electrical conductivity and high elasticity make it ideal for our application. When the insole is pressed, the distance between the two conductive plates decreases, leading to a change in the measured capacitance value. The more pressure (or force) is applied to the top layer of the insole, the greater the capacitance variation is, but how to relate these is not straightforward. Determining this relationship experimentally and validating it are primary purposes of this paper.

### 2.2. Physics-Based Approach

When calibrating the taxels, the goal is to find a relationship that links the behavior of the sensor, in terms of capacitance variation, to the evolution of the input pressure value. In other words, the measurement output (i.e., the capacitance values) has to be used to estimate the input pressure that caused it:(1)$$\hat{P}=f\left(C\right)$$

The expression above is very general and, in principle, completely determined by the physical parameters of the system. In the following, however, we show why resorting to identification of these parameters is not effective, motivating the different approach used here involving use of a polynomial model for the taxel.

The general expression used to evaluate the analog capacitance created between two conductive plates is:(2)$$C_{a}=\frac{\epsilon_{r}\epsilon_{0}A}{d}$$
where $\epsilon_{0}$ and $\epsilon_{r}$ are, respectively, the free space and (relative) dielectric material permittivity, *A* is the area of the taxel and *d* is the distance between the two capacitor plates [26]. The mechanical elastic behaviour of the dielectric can be approximated with a set of parallel springs connecting the two plates by Hooke’s law [27]:(3)$$F=k(d_{0}-d)$$
where *k* is the stiffness of the material and $d_{0}$ is the distance between the plates at rest. Since the taxel considered is very small (A≈19 mm^2^), it is possible to assume uniform pressure *P* acting on it, so that:(4)F=AP

By substituting (Equation 3) and (Equation 4) in (Equation 2) and solving for *P*, the pressure estimate for the taxel can be found:(5)$$P=\frac{kd_{0}}{A}-\frac{k\epsilon_{r}\epsilon_{0}}{C_{a}}$$
with some caveats:*k* changes with the pressure and with time, since the linear model is valid only for small deformations;$d_{0}$ is not constant due to mechanical hysteresis;*A* can vary slightly from one taxel to another.

In our model, we deal with the first two caveats, since the taxel area is well-defined by the production process up to the $O(10^{-4})$ level. In addition to this, it should be noted that the relationship in (Equation 2) is an approximation, valid only when the aspect ratio of the capacitor is small, while the sensors that we consider might fall outside the scope of this hypothesis. Hence, it should be substituted by more accurate modeling of our situation, such as that proposed in [28,29,30] for circular capacitors, but at the cost of more complex formulation and an increased number of (possibly time-varying) parameters to be estimated. Ultimately, this kind of approach does not seem to be a viable option, so we switch to a parameterisation of the system for the purpose of achieving a practical calibration.

### 2.3. Problem Statement

Having discarded the possibility of modeling the responses of the taxels by leveraging a purely physical description of their behaviour, we need to find a different option to relate the pressure acting on the insoles with the outputs of the sensors embedded in it. Such a relationship is, instead, approximated by means of a polynomial model of the taxels, since this appears to offer a reasonable representation of the underlying physics, as explained below.

We select the best model by applying a two-step identification procedure. We first assume that the sensor’s behaviour depends just on its current state (i.e., pressure applied and voltage output), exploring how models of different complexity perform, and we adopt a reasonable trade-off between the order of the polynomial and the accuracy of the resulting pressure estimate. Then, we expand this model, introducing the past values of the taxel’s output to refine the pressure estimate by taking its dynamics into account. Once the general structure of the model is known, each taxel on the insoles is calibrated independently from the others.

We first provide a detailed description of the calibration setup that was employed in our study since the experimental data obtained is the basis on which the model identification is performed.

## 3. Calibration Setup

A data-driven approach is taken to identify the parameters of our model; therefore, the experimental setup that is used should be able to deliver meaningful data in real-time. Because our insoles are imbued with sensors, an efficient solution to the calibration problem needs to treat all of them at the same time. The main idea is to perform an experiment during which the same pressure is applied over *all* the taxels. This leads to a faster, more scalable procedure for virtually calibrating different kinds of sensor arrays used in other applications. As mentioned in Section 1.2, refs. [6,20,21] have taken first steps in this direction by using an air-tight system for which the main drawback is the limited pressure range achieved. To overcome this constraint, a different setup was developed, the main idea of which was to use a pressurized chamber to produce higher pressures. It was composed of:a steel pressure tank, capable of withstanding a maximum pressure of 10 bars;a custom support, specially designed to house the insole during calibration;a 24-L air compressor;a closed-loop pressure regulator (QB4 from ProportionAir);a high-precision pressure sensor (PN2014 from IFM);several high-pressure elements (pipes and valves) to connect the various parts of the setup together.

Figure 1 shows a cross-sectional view of the tank-support system: the insole is first secured inside the mobile support, which is manufactured to hold both the insole and its electronics safely, and then inserted into the tank, allowing wired communication between the insole and the user’s laptop for real-time data collection. As a result of use of a plastic membrane placed in the top part of the calibration support, the insole always experiences differential pressure $P_{tank}-P_{atm}$, while being in a realistic atmospheric-pressure environment. This is beneficial for cases in which different kinds of dielectric materials are considered (e.g., plastic foams or other materials containing air, since they may behave differently if tested in an environment that is not at atmospheric pressure). Different sensor arrays of various shapes can be calibrated in this device, provided that an appropriate support is designed.

Outside of the tank, a standard LPC1768 electronic board is used to control the behaviour and set the reference for the pressure regulator via customized firmware, which allows the user to interact in real-time with the system through a laptop. A voltage amplifier circuit is implemented to scale the output of the board (0–3 V) up to the expected command signal range for the pressure regulator (0–10 V). Conversely, a voltage divider performs the opposite voltage conversion, enabling the pressure sensor to be read directly from the same board.

Using this setup (depicted schematically in Figure 2), the dataset needed for the calibration of a single insole can be collected in no more than 15 min and can be fed to our model identification procedure.

## 4. Modeling and Identification

This section presents the contributions of this paper in terms of the modeling strategies chosen for the taxel’s characteristics, together with the algorithms deployed for its identification. Details about the data collection and filtering are also discussed.

### 4.1. Polynomial Model

We considered the use of a general polynomial expression of the following form, written for taxel *i*:(6)$${\hat{P}}_{i}\left(C_{i},t\right)=\pi \left(C_{i,t}\right)+\sum_{k=1}^{n_{s}}h_{k}\left(C_{i,t-k}\right)$$
where

$C_{i}$ is the vector containing all of the capacitance measurements recorded by taxel *i*;${\hat{P}}_{i}\left(C_{i},t\right)$ is the pressure estimated by taxel *i* at time *t* based on its measurements;$\pi \left(C_{i,t}\right)=a_{0,0}+a_{0,1}C_{i,t}+...+a_{0,n_{p}}C_{i,t}^{n_{p}}$ is a polynomial of order $n_{p}$ that links the instantaneous value of the capacitance $C_{i,t}$ to the pressure estimate;$h_{k}\left(C_{i,t-k}\right)=a_{k,1}C_{i,t-k}+...+a_{k,n_{ps}}C_{i,t-k}^{n_{ps}}$ is a polynomial of order $n_{ps}$ contributing to enhance the pressure estimate with information coming from the past state of the sensor (at time t−k);$n_{s}$ is a parameter selecting the number of past samples to be considered for the current estimate.

In the proposed approach, the orders of the various polynomials are part of the parameter set found via optimization. We first identify an appropriate range of the polynomial order $n_{p}$ using a reduced model ${\hat{P}}_{i,R}\left(C_{i},t\right)=\pi \left(C_{i,t}\right)$, and then we perform a broader optimization to retrieve the optimal combination of $n_{p}$, $n_{s}$ and $n_{ps}$, together with all the other coefficients $a_{k,j}$ of the polynomials involved in (Equation 6). With this procedure, we aim to generalize the approach found in many related investigations, by conceptually first selecting a reduced model that is already somewhat representative of the underlying physics and then expanding it with information on the past history of the taxel to improve it.

### 4.2. Data Collection

Several calibration experiments were performed with the setup presented in Section 3, during which we simultaneously recorded capacitance data coming from the insole together with the real pressure that was being applied on it. When the uncompressed insole undergoes sudden pressure variations, the initial effect of the hysteresis and elasticity of the dielectric material could be relevant but difficult to take into account. For this reason, before starting data recording, we conditioned the insole by performing several pressure cycles in the tank, accepting that our identified model would possibly be less accurate in the first few seconds of use than in a real scenario (e.g., when a person who wears the insole immediately walks on it), but aiming atimproved accuracy once this early transient is over.

The calibration datasets were acquired to account both for the dynamic and static response of the sensors. The pressure inside the tank was increased with steps of 0.2 bar between $P_{tank,min}=1$ bar and $P_{tank,max}=4.6$ bar, roughly every 10 s. For each step signals were sampled both in the proximity of the transients and after the system settled to the steady state in ≈2.5 s. The two datasets comprised 25% and 75% of the full ensemble, respectively, and were recorded at 50 Hz. The insole underwent a differential pressure in the range of 0–3.6 bar that matched most of the pressure range that would be encountered in everyday use. Starting from the relationship between the pressure acting on the insole, the mass *m* of the person and the surface that is being compressed are:(7)$$P=\frac{mg}{A\cdot n_{t}}$$
where *g* is the gravitational acceleration, $A=3.24\times 10^{-4}$ m^2^ is the area of a single triangular patch and $n_{t}$ is the number of patches on which the force is exerted. A subject of 80 kg balancing on one foot (28 triangles) would produce an average pressure P≈0.85 bar, while standing on the toes would result in P≈3.4 bar (considering just seven triangles activated).

### 4.3. Mathematical Methods

In order to retrieve the most descriptive model for the taxels from the model class introduced in (Equation 6), an identification technique must be implemented. The various parameters are estimated by means of a least-squares optimization algorithm. Using matrix notation, we can rewrite (Equation 6) more compactly as
(8)$${\hat{P}}_{i}\left(C_{i},t\right)=\phi_{t}{\left(C_{i}\right)}^{T}\lambda_{i}$$
where $\phi_{t}\left(C_{i}\right)\in R^{n_{p}+n_{s}\times n_{ps}+1}$ is the regressor of the experimental data built at time *t* for the *i*th-taxel, and $\lambda_{i}\in R^{n_{p}+n_{s}\times n_{ps}+1}$ represents the coefficient vector for that specific taxel. They can be explicitly written as:(9)$$\phi_{t}\left(C_{i}\right)={\left[1\;C_{i,t}\;\ldots \;C_{i,t}^{n_{p}}\;C_{i,t-1}\;\ldots \;C_{i,t\ldots 1}^{n_{ps}}\;\ldots \;C_{i,t-n_{s}}\;\ldots \;C_{i,t-n_{s}}^{n_{ps}}\right]}^{T}$$
(10)$$\lambda_{i}={\left[a_{0,0}\;a_{0,1}\;\ldots \;a_{0,n_{p}}\;a_{1,1}\;\ldots \;a_{1,n_{ps}}\;\ldots \;a_{n_{s},1}\;\ldots \;a_{n_{s},n_{ps}}\right]}^{T}$$

Since the number *d* of samples acquired in the calibration experiment is much larger than the number of unknown coefficients, it is possible to define an over-constrained system of the following form:(11)$$P=\Phi_{i}\lambda_{i}$$
where $P\in R^{d}$ is the vector containing the ground-truth pressure inside the tank (measured with the pressure sensor), and $\Phi_{i}\in R^{d,n_{p}+n_{s}\times n_{ps}+1}$ is the regression matrix for the taxel defined as:(12)$$\Phi_{i}=\left[\begin{array}{c} \phi_{1}{\left(C_{i}\right)}^{T}\\ \phi_{2}{\left(C_{i}\right)}^{T}\\ \ldots \\ \phi_{d}{\left(C_{i}\right)}^{T} \end{array}\right]$$

Note that an initial padding of $n_{s}$ elements in $C_{i}$ is required for consistency.

The over-determined system in (Equation 11) can be solved by means of a least-squares approach to find the optimal value for $\lambda_{i}$:(13)$$\lambda_{i}^{★}=\underset{\lambda_{i}}{\mathrm{argmin}}\left|\right|\Phi_{i}\lambda_{i}-P{\left|\right|}^{2}$$

Resorting to the quadratic programming formulation, we can rewrite (Equation 13) as
(14)$$\lambda_{i}^{★}=\underset{\lambda_{i}}{\mathrm{argmin}}\left(\frac{1}{2}\lambda_{i}^{T}H\lambda_{i}+\lambda_{i}^{T}g\right)$$
where the Hessian matrix is defined as $H=\Phi_{i}^{T}\Phi_{i}$ and the gradient is $g=-\Phi_{i}^{T}P$. Feeding this problem to conventional off-the-shelf optimizer (OSQP [31]) enables the optimal values of the parameters to be found defining the physical model of taxel *i*. The same procedure is leveraged to set up a unique optimization problem for each of the taxels on the insole, since their pressure response can vary from one sensor to another. Notably, this is a relevant theoretical difference with respect to [6], where a single optimization problem was considered for all of the taxels together, leading to the presence of non-physically meaningful mixed terms in the hessian and to much higher computational complexity.

### 4.4. Data Preprocessing

Even if particular care is used while collecting and recording the datasets, some preprocessing is still required to filter out noise and to achieve a better calibration. Since we recorded the data at 50 Hz, we know that excessive variations of the pressure value are not possible from one sample to the next. Therefore, we discarded the samples whose difference from the previous value exceeded a threshold (heuristically set to 0.05 bar). The resulting datasets were also filtered with an exponential filter to decrease the noise level.

A final step is necessary: when looking at (Equation 13), it becomes evident that the accuracy of the solution depends on the condition number of $\Phi_{i}$. To avoid an ill-conditioned problem, we:implement a Tikhonov regularization [32] of our problem to favour solutions with a lower norm;normalize the regressor $\Phi_{i}$ (i.e., each column is scaled by the value of its maximum element). As a result, all the elements in the new regressor will belong to the range [0,1]. This is necessary, since, from its definition in (Equation 9), it is evident that different columns of $\Phi_{i}$ have very different orders of magnitude. The solution of the scaled problem is also to be scaled similarly in order to counter-balance the smaller value range considered;subsample the original dataset to scale down the dimension of the problem by retaining a similar level of information.

Eventually, the scaled problem to be addressed results in
(15)$${\lambda_{i}^{\prime }}^{★}=\underset{\lambda_{i}^{\prime }}{\mathrm{argmin}}\left|\right|\Phi_{i}^{\prime }\lambda_{i}^{\prime }-{P||}^{2}+k||\lambda_{i}^{\prime }{\left|\right|}^{2}$$
where $\Phi_{i}^{\prime }$ is the scaled regressor and *k* is the Tikhonov regularization term (set to $10^{-3}$). The optimal solution $\lambda_{i}^{★}$ is retrieved from ${\lambda_{i}^{\prime }}^{★}$ by dividing all of its elements by the same scaling factors used for the calculation of $\Phi_{i}^{\prime }$.

### 4.5. Calibration of the Base *Instantaneous* Polynomial

As anticipated in Section 4.1, first a reduced model is identified, considering the pressure-capacitance relationship to be instantaneous (i.e., just present values of the capacitance $C_{i}$ are involved in the estimation of ${\hat{P}}_{i,R}\left(C_{i},t\right)$). The polynomial order $n_{p}$ of such a model is varied in the interval [1:25], assumed to be wide enough to represent the dynamic evolution of the state of the sensor. Using the calibration datasets described in Section 4.2, the optimal coefficients corresponding to each value of $n_{p}$ are retrieved. The performances of the different models are evaluated in terms of root-mean-squared error (RMSE) by comparing the pressure estimates of each model to a ground truth validation dataset, general enough to cover pressure steps (as in calibration), but also more dynamic/smoother variations (see Section 5).

As is clear from Figure 3, $n_{p}=7$ grants a local minimum and better performances can be achieved with much higher orders. In view of the trade-off between model complexity and accuracy of the results, the previous observation justifies considering only the interval [1:8] for the parameter $n_{p}$ in the broader optimization that is carried out below.

### 4.6. Expansion of the Base Model

Since each taxel is a physical system composed of deformable elements subject to hysteresis and non-linearities, its response to an input pressure at a certain time instant must somewhat depend on the past states of the system itself. By considering the full expression in (Equation 8), we can introduce this dependency on the history of the system in an optimal way. Since the investigation in Section 4.5 suggested to set $n_{p}\leq 8$, the same boundary is extended naturally to $n_{ps}$, which has a similar physical meaning. In this way, the dimensions of the regression matrix remain acceptable and the range of parameters explored is at the same time meaningful and not too wide.

The problem as formulated in (Equation 15) is solved for $n_{p}\in \left\{1,2,3,4,5,6,7,8\right\}$, $n_{ps}\in \left\{1,2,3,4,5,6,7,8\right\}$ and $n_{s}\in${0,10,20,30,40,50,60} (and all the possible combinations of these). The same training and validation sets described previously were used. The metric to compare the different models was again the RMSE of the estimated pressure of each model against the pressure recorded in the validation dataset. An overview of the results of this broad optimization is shown in Figure 4, where it is apparent that a more complex model gives generally better results. In particular, the larger the number of past samples $n_{s}$ considered, the more accurate the pressure estimate becomes. The same is true for an increasing value of $n_{ps}$.

To establish a trade-off between accuracy and model complexity, it is reasonable to favour lower-order models which are capable of estimating the external pressure with errors that are almost negligibly worse than those coming from the best, much more convoluted, models.

The numerical results (see Figure 5) show that, by increasing the value of $n_{ps}$, at first the models perform better, while beyond $n_{ps}=4$ the estimation RMSE increases slightly, to decrease again only considering $n_{ps}\geq 7$. Recalling that $n_{ps}$ represents the order of each of the polynomials featuring the past capacitance values (as defined in (Equation 6)), it is clear that the value of such a parameter has a great impact on the dimension of the optimization vector $\lambda_{i}\in R^{n_{p}+n_{s}\times n_{ps}+1}$. Therefore, the value of $n_{ps}=4$ was chosen to reduce, as much as possible, the search space for the optimization algorithm, despite a slightly higher error.

## 5. Results and Discussion

Our identification procedure highlights two models in particular that can be considered to be satisfactory. In light of the considerations explained in the previous sections, we sought to find a model for the capacitive taxel that was able to accurately estimate the input pressure without being disproportionately complex. Therefore, in the case of equivalent performances, the simpler model should be chosen. We compare the performances of the optimized models in terms of their overall RMSE in validation. Since each taxel shares with the others the same model structure but is calibrated independently, it delivers a unique estimate for the pressure it is experiencing. For conciseness, the RMSE that we consider here is averaged over all the pressure estimates delivered by the taxels (some taxels presented corrupted measurements and were excluded from this evaluation).

The analyses presented in Figure 6 and Table 1 show two possibilities for the optimal model of the taxels:**Model A**, found by setting $\left\{n_{p},n_{s},n_{ps}\right\}=\left\{3,60,4\right\}$, that achieves a lower RMSE on the validation scenarios considered;**Model B**, found by setting $\left\{n_{p},n_{s},n_{ps}\right\}=\left\{3,40,4\right\}$, that, even if slightly less accurate, requires a lower dimensionality of the optimization vector.

While granting a lower RMSE, the first solution is less reactive to variations of the input pressure (since it accounts for a longer history of the system state). In certain applications (for example, when it is needed to collect and analyse dynamic gaits), the second solution may be preferred as it decreases the response time of the system. It is interesting to note that both these models are characterized by a base instantaneous polynomial of order $n_{p}=3$ and account for the past samples by using polynomials of order $n_{ps}=4$. In Table 1, we also show the results for another optimized model that we call *Model C*. This model is much more complex than the two discussed above as can be seen by the much higher number of parameters that need to be optimized (488 against roughly 200 for the other two models). While we show that this very cumbersome model hints at achieving a slightly lower validation error, we argue that its additional complexity has significant disadvantages in terms of memory requirements and the reactivity of the model, so we do not use it.

The two optimal models presented above have very interesting parallels with the results of a previous investigation reported in [6] (concerned with the same insoles), where a pure polynomial of order three was selected and employed to model the taxel’s behaviour. In this sense, the increased flexibility of our model, that also considers past states of the taxel, results in a much more accurate estimation for the pressure value acting on the taxels themselves at each time instant (see Table 1). To report these results, the instantaneous model proposed in [6] was calibrated and validated again, using the same datasets considered in our identification procedure. Including the past state successfully confirmed our initial claims and improved the pressure estimates delivered by the model by halving the overall RMSE on the validation dataset considered (the RMSE was reduced from almost 9% to about 4.4% of the pressure range considered). Notably, the results obtained were not only more precise, but were also obtained much more quickly owing to the different formulation of the optimization problem, as pointed out in Section 4.3. Thus, the presented method not only has a stronger physical base, but uses an optimization which is much faster to set up and solve. In comparison to the algorithm suggested in [6] our algorithm proved to be 100 times faster (∼3 s against 260 s, on a portable Intel i7-11 laptop with 16 Gb RAM). To conclude, we show in Figure 7 the pressure estimates achieved by one of the proposed models (*Model A*) on the validation dataset considered. The curves corresponding to the pressure estimates of two different taxels are shown, namely the taxel associated with the lowest and highest RMSE. Although the more accurate estimation is clearly an improvement, the other model is capable of reconstructing the input pressure quite well.

## 6. Conclusions

We presented a complete pipeline for calibrating a dense array of capacitive pressure sensors, constituting a pair of sensorized insoles. Using a rich experimental dataset collected via a new calibration setup, a polynomial model incorporating past samples from the taxels was proposed. This model enables estimation of the pressure acting on each single taxel of the insoles and copes effectively with the non-linearities that are characteristic of the physical system. This estimation can be performed in real-time once the model has been identified. This allows us to use data coming from the insoles as a basis for developing further algorithms, enabling online monitoring of the status of a person. Future work will include comparison of the estimation achieved by the insoles when worn by a user with that obtained using classical methods (e.g., force plates) and use of our insoles equipped with suitable onboard computing devices in real-life scenarios.

## Figures and Tables

**Figure 1 sensors-23-04501-f001:**
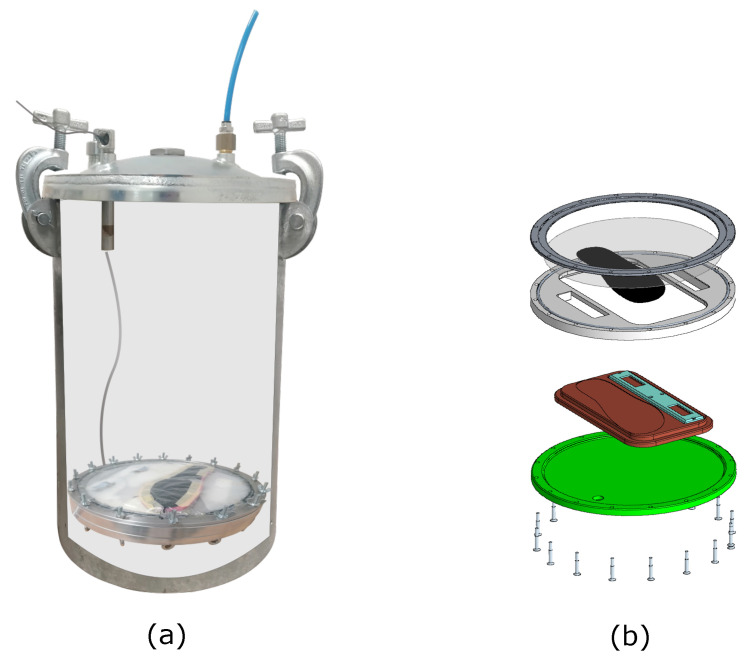
The novel calibration setup that we propose is visualized: (**a**) cross-section of the pressure tank, enabling visualization of the support holding the insole during the calibration phase; (**b**) CAD image of the parts composing the support, safely housing the insole and its electronics.

**Figure 2 sensors-23-04501-f002:**
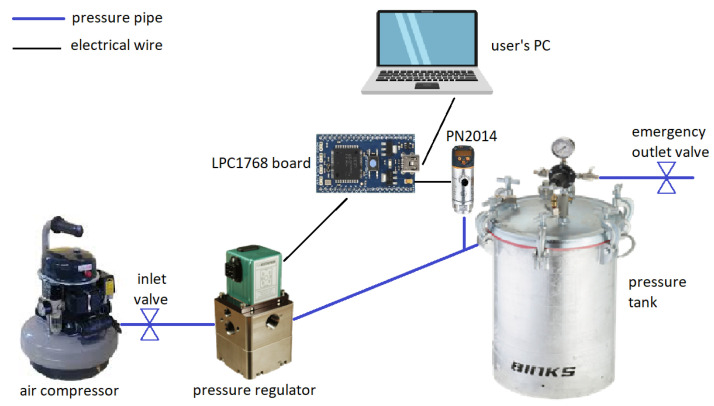
Schematics of the calibration setup.

**Figure 3 sensors-23-04501-f003:**
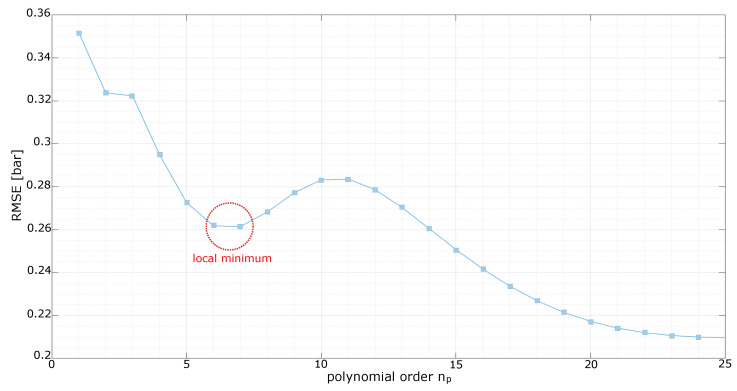
Calibration of the base instantaneous polynomial, showing the RMSE on the validation dataset as a function of the order $n_{p}$ of the polynomial. A very large range of values for $n_{p}$ is examined; however, from the plot it is evident that a local minimum can be found at $n_{p}=7$. Therefore, a restricted yet meaningful range for the polynomial orders (up to $n_{p}=8$) is considered in the second step of the calibration.

**Figure 4 sensors-23-04501-f004:**
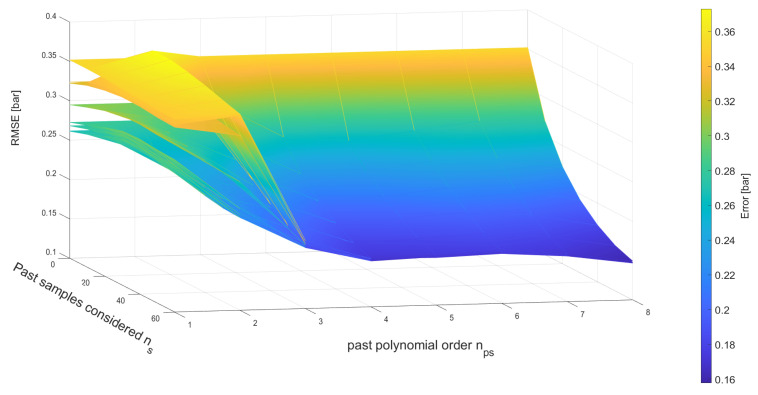
RMSE resulting from the pressure estimates achieved with the optimal parameter sets $\lambda_{i}$, depending on various values of $n_{p}$, $n_{s}$ and $n_{ps}$. Each surface represents a different value of $n_{p}$, as described in Section 4.6.

**Figure 5 sensors-23-04501-f005:**
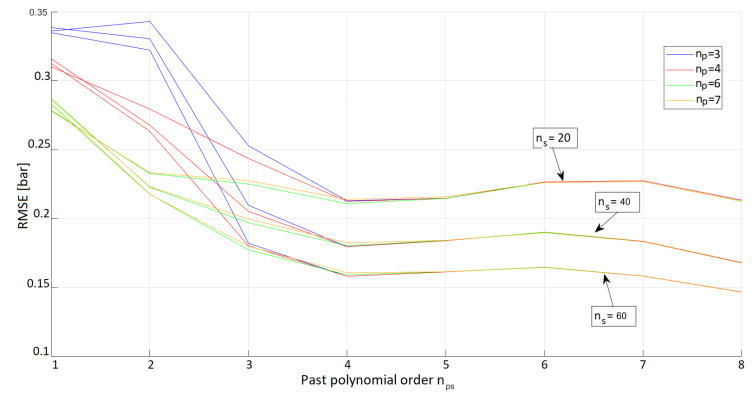
Comparison of the different performances, in terms of RMSE, for some chosen models. Models with just 20, 40 and 60 past samples are analyzed, characterized by base instantaneous polynomials of order 3, 4, 6 or 7.

**Figure 6 sensors-23-04501-f006:**
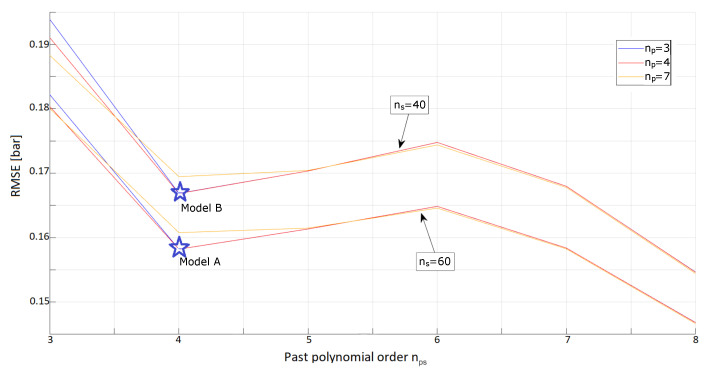
Analysis of the best results achieved. Of the different models represented here, two are selected by virtue of their accuracy and reduced complexity, and indicated with a star-shape on the graph. **Model A** represents the best trade-off in the validation scenarios considered, since it achieves a smaller RMSE when compared to other models which are still easily treatable (i.e., where $n_{ps}\leq 7)$. **Model B** retains good performances (as well as the property of being locally optimal given the number of past samples considered) and is more reactive to dynamic variations of the pressure signal as it considers a reduced range of the dynamic history of the sensor’s state.

**Figure 7 sensors-23-04501-f007:**
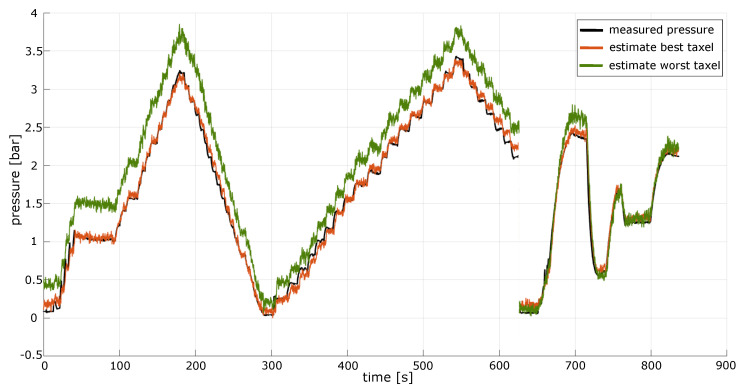
Pressure estimates retrieved from *Model A* on the validation dataset (comprised of 3 different experiments). The true pressure level recorded during the experiments is shown, together with the best-fitting estimate (orange), and the worst-fitting estimate (green). Both curves show high similarity to the target of the estimation.

**Table 1 sensors-23-04501-t001:** Comparison between different models in validation.

	RMSE [bar]	$n_{p}$	$n_{s}$	$n_{\mathrm{ps}}$	Dimension of $\lambda_{i}$	Error %
Model A	0.158	3	60	4	244	4.4
Model B	0.167	3	40	4	164	4.6
Model C	0.147	7	60	8	488	4.1
from [6]	0.322	3	0	0	3	8.9

## Data Availability

The data and code used in this study are freely available at https://github.com/ami-iit/paper_belli_2023_sensors_insole-calibration.

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
