# Peer review of "Modeling and Calibration of Pressure-Sensing Insoles via a New Plenum-Based Chamber"

_sensors, 2023, doi:10.3390/s23094501_

Round 1

Reviewer 1 Report

This manuscript presents a customized calibration setup and procedures for wearable insoles with 280 capacitive pressure sensors. The authors propose a refined calibration procedure by leveraging a polynomial model for high dense sensor arrays in insole. The paper is completed in the present form, but it is a bit lengthy. Some results can be attached as supporting materials. The reviewer recommends publishing it in the journal.

Author Response

Dear reviewer 1, 

we would like to thank you for your nice words regarding our work. We answer in the following:

  1. following comments from other reviewers, we have slightly modified and cut section 4.3. This reduced the overall length of the paper, and from our side we feel that the results as they are presented now allow for a comparison between various identified models at a glance. Moreover, the paper's length complies with the recommendation for publication in Sensors MDPI. Hence, we would opt for keeping the current structure of the result section.

Best regards,

the authors

Reviewer 2 Report

The manuscript aims to present a calibration method for pressure sensing insole, which is interesting for wearable clothing. It adopts nonlinear mathematical model to acquire and analyze pressure signal for solving the current problem of nonlinear sensor signal. The model is proper and there has good accordance between theoretical and experimental results. I think that the manuscript can be accepted. 

Author Response

Dear reviewer 2, 

we would like to thank you for your nice words regarding our work. We answer in the following:

  1. the indication that the paper requires "Extensive editing of English language and style" seems to us the consequence of an error when selecting the correct box to tick. We are fairly confident that the language we used is clear and accessible to the community. We have checked our manuscript once more and possibly adjusted spellings that were inserted erroneously.

Best regards,

the authors

Reviewer 3 Report

Improve the description of the formulas and of the methods in 4.3 !

Make figures more readable (size of police) and when appropriate with average results: comment on the standard deviation or show error bars !

Author Response

Dear reviewer 3, 

we would like to thank you for your careful review. We answer in the following:

  1. we have improved section 4.3 clarifying our notation, and cutting off some details which were probably a bit overwhelming for the readers. Also section 4.1 has been changed slightly accordingly
  2. we have increased the size of the fonts in the images. Unfortunately, because of a change in the setup that was used for the experiments (and relocation of some authors) we cannot perform the experiments that would be needed to add the requested information about the standard deviation of the results.

We will keep your valuable opinion in mind in our future works.

Best regards,

the authors

Reviewer 4 Report

The paper has a rigorous structure, clear argumentation, and fully explains the working principle of experimental instruments and experimental results. There are several minor problems in the paper:

1. The text in Figure 7 is much smaller than the others

2. The ordinate text in Figure 3 and Figure 6 is not the same size

Author Response

Dear reviewer 4,

we would like to thank you for your nice words regarding our work. We answer in the following:

  1. we have increased the size of the fonts in the images, and modified the labels as accurately noted by your review.

Best regards,

the authors